# A Challenge of COVID—19: Associated Infective Endocarditis with *Streptococcus gordonii* in a Young Immunocompetent Patient

**DOI:** 10.3390/medicina57121298

**Published:** 2021-11-26

**Authors:** Manuela Arbune, Alina-Viorica Iancu, Gabriela Lupasteanu, Mihaela-Camelia Vasile, Victorita Stefanescu

**Affiliations:** 1Clinical Department, Medicine and Pharmacy Faculty, Dunarea de Jos University of Galati, 800008 Galati, Romania; mihaela.vasile@ugal.ro; 2Morphological and Functional Science Department, Medicine and Pharmacy Faculty, Dunarea de Jos University of Galati, 800008 Galati, Romania; alina.iancu@ugal.ro (A.-V.I.); victorita.stefanescu@ugal.ro (V.S.); 3Doctoral School, Ovidius University of Constanta, 800179 Galati, Romania

**Keywords:** COVID-19, *Streptococcus gordonii*, infectious endocarditis, oral microbiome

## Abstract

The COVID-19 pandemic is a new challenge for the diagnosis and treatment of infective endocarditis (IE). Fever and other unspecific symptoms of coronaviral infection could be misleading or masking its manifestations. We present the case of a young patient admitted for persistent fever, profuse sweating, headache, articular pain, myalgias, and weight loss. She reported regression taste and smell disorders compared to a month earlier when diagnosed with moderate COVID-19 pneumonia. While the RT-PCR SARS-COV-2 test was positive, she was admitted to a COVID-19 ward. Investigations of febrile syndrome revealed two positive blood cultures with *Streptococcus gordonii* and the presence of vegetations on the aortic valve, supporting a certain diagnosis of IE. After six weeks of antibiotic treatment, the patient had clinical and biologic favorable outcomes. *Streptococcus gordonii* is a common commensal related to the dental biofilm, although there were no caries in our patient. The influence of COVID-19 infection on the human microbiome by modifying the virulence of some commensal germs may be a risk factor for IE pathogenesis on native valves and requires the vigilance of clinicians for suspicion of this disease.

## 1. Introduction

The infection COVID-19 was initially defined as a respiratory viral disease. However, the identification of viral pathophysiologic mechanisms and the clinical experience gained by healthcare care of patients with COVID-19 have classified it as a multisystemic disease [1].

Among the extra-respiratory manifestations, cardiovascular injury has a wide variety of clinical forms, such as myocarditis, pericarditis, myocardial infarction, vasculitis, venous, or arterial thromboembolism, pulmonary thromboembolism, and acute heart failure. Infective endocarditis (IE) related to COVID-19 has been reported in a small number of patients, with a possible contribution of a severe inflammatory response, endothelial dysfunction, hypercoagulability, or immunosuppression of some drugs used to treat this infection [2,3].

## 2. Case Presentation

A 23-year-old female student presented to the Infectious Diseases Clinic in January 2021, during the context of the COVID-19 pandemic, with fever (38.5 °C), asthenia, intense sweating, myalgias, arthralgia, frontotemporal headache, pain in the left lower extremity, and weight loss (8% in the last six weeks). Family history includes the grandfather with stroke, the mother with hypertension, and obesity. The personal history mentions frequent streptococcal angina during childhood and adolescence and benzathine penicillin G treatments, but medical documentation was not available. The patient denies smoking, alcohol, and drugs use. She has not been vaccinated against pneumococcal diseases, influenza, or COVID-19 and denies contact with sick people.

The onset of the disease was almost a month ago, with headache, cough, fever, taste, and odor disorders. She presented to the emergency department of the county hospital, and the rapid test for SARS-Cov-2 was positive. Considering the good general condition, normal respiratory function, and normal biological data, the case was classified as mild form, and it was recommended to isolate and receive treatment at home, with oral cefuroxime and non-steroidal anti-inflammatory drugs. Although the initial symptoms improved, the patient returned to the emergency department for retrosternal pain, but no signs of aggravation were found, continuing home care. After three weeks from the onset, the malaise and fever over 38 °C returned, and she had requested a new emergency consultation. The RT-SARS-Cov-2 test was still positive, and she was hospitalized in a COVID-19 department.

The objective examination reported fever 38.5 °C, oxygen saturation (SO_2_) 98% on room air, respiratory rate (RR) 20/min, blood pressure (BP) 120/80 mmHg, heart rate (HR) 113/min, grade II aortic systolic murmur, bilateral rales on auscultation of the lung, sweaty skin, without rash, unremarkable otherwise. The chest X-ray image showed increased interstitial markings, and laboratory investigations revealed leukocytosis, neutrophilia, normocytic normochromic mild anemia, inflammatory syndrome, reacted D-dimers, and microscopic hematuria (Appendix A). Antibiotics with Ceftriaxone and prophylactic anticoagulation were provided. Two sets of blood cultures collected at admission evidenced positive results for *Streptococcus gordonii*. The COVID-19 RT-PCR test was negative after two weeks of hospitalization. Transthoracic cardiac ultrasound found a mobile structure of 0.5 cm on the ventricular side of the aortic cusp, suggesting vegetation (Figure A1). 

Thus, in a young patient with COVID-19 infection, two major criteria were identified for the diagnosis of IE, according to the modified Duke’s definition [4]. Antibiotic therapy was decided according to the European Society of Cardiology Guide, continuing Ceftriaxone for six weeks [5]. The fever decreased to 72 h and no other pathological changes were found during hospitalization, except for the stationary aortic murmur. Subjectively, she developed self-limiting pain in the left lower limb and anxiety disorder, but the evolution was favorable.

The aortic vegetation highlighted by the transthoracic ultrasound examination recovered completely, although aortic and mitral regurgitation still occurred. A cytolytic syndrome appeared in the fourth week of treatment but returned to normal after the end of antibiotic treatment. The cytolytic syndrome could imply toxic drug hepatitis, a rare adverse reaction to Ceftriaxone caused by cholestatic, hepatocellular, or combined lesions [6]. The other biological parameters were gradually normalized.

Oral examination with panoramic radiographs was normal, just a partially erupted wisdom tooth was observed.

## 3. Discussion

*Streptococcus gordonii* belongs to the *Streptococcus sanguinis* group, classified as viridans streptococci, along with the groups *S. mutans*, *S. mitis*, *S. anginosus*, *S. salivarius*, and *S. bovis* [6]. *Streptococcus gordonii* is considered a commensal bacterium found in soil, water, and the composition of a part of the oral, cutaneous, and intestinal microbiome [7]. Peculiarly, *Streptococcus gordonii* can adhere to the dental surface, forming a biofilm that modulates the pathogenicity of secondary colonizer bacteria through specific communication mechanisms [7]. 

However, *Streptococcus gordonii* can be an opportunistic bacterium associated with localized infections, such as apical periodontitis or disseminated infections after bacteremia, causing secondary foci, such as endocarditis, empyema, perihepatic abscesses, pyogenic spondylitis, or spondylodiscitis [8]. The influence of COVID-19 infection on the human microbiome and the favorable role in modifying the virulence of some commensal germs may be a risk factor for the pathogenesis of IE in native valves, which requires the vigilance of clinicians for suspicion of this disease [9,10].

The medical history of our case mentioned recurrent pharyngitis, probable with streptococcal etiology, although the patient was not diagnosed with renal complications. This is in accordance with the small incidence of poststreptococcal glomerulonephritis in Europe. The clinical presentation is variable, from asymptomatic to nephritic syndrome or even severe kidney failure [11]. Kidney involvement in the IE was considered primary embolic, while current medical reports support that 80% of cases represent focal, segmental, or diffuse proliferative glomerulonephritis, confirmed in autopsy studies [12]. 

Bacterial endocarditis associated with COVID-19 infection is rare. A multicenter study conducted by the European Society of Cardiology, which included 3011 patients with COVID-19, found an incidence of COVID-19-associated heart complications of 11.6%, but IE was reported in only 0.1% of cases [13]. A recent systematic review identified 15 cases of IE in the context of COVID-19 in patients aged 20 to 73 years, especially men, most cases of native valves, aortic position, and predominated etiology with *Enterococcus faecalis* [2]. Other confirmed etiologies were *Methicillin-sensitive* or *Methicillin-resistant Staphylococcus* spp., *Pseudomonas aeruginosa*, *Candida albicans*, but no case of *Streptococcus gordonii* was found [14,15,16,17]. The causative relationship of COVID-19 and IE was not proved until now and could be coincidental in the context of health care deficient organization during the pandemic [18].

The particularities of the case presented are the appearance of IE in a young immunocompetent woman, in the native valve without known previous lesions, the rare etiology of bacterial endocarditis with *Streptococcus gordonii*, and the context of diagnostic during the second pandemic wave COVID-19. The common source of bacteremia with *Streptococcus gordonii* is a periodontal lesion, but it has not been identified in the present case. However, the source may be the oral cavity, possibly at the level of the wisdom tooth that had a discrete gingival inflammation, highlighted by imaging after discharge from the hospital. The diagnostic difficulty of the case was related to the overlap of symptoms of COVID-19 infection and infective endocarditis.

## 4. Conclusions

The presented case was a moderate form of COVID-19 and concomitant endocarditis with a favorable course. Although complete remission of vegetation was notified, careful follow-up is considered for possible late complications. Changes of the oral microbioma during COVID-19 disease is a favoring factor implied in bacterial complications. The overlapping symptoms of COVID-19 infection can mask the course of other severe infections, such as endocarditis. The influence of COVID-19 infection on specific pathogenic mechanisms of infectious endocarditis is not clearly specified, and further studies are required.

## Data Availability

All the current data is available on request from the authors.

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
