# Peer review of "A Challenge of COVID—19: Associated Infective Endocarditis with Streptococcus gordonii in a Young Immunocompetent Patient"

_medicina, 2021, doi:10.3390/medicina57121298_

Round 1

Reviewer 1 Report

Present article ''A Challenge of COVID - 19: Associated Infectious Endocarditis with Streptococcus gordonii in a young immunocompetent patient'' consists another publication regarding co-infection of COVID-19 with infective endocarditis in an otherwise immunocompetent adult. The case report is well structured, fluent, highlighting the important points.

Few comments are listed below:  

  1. Please provide specification of SARS-COV2 mode of testing (e.g line 20)
  2. Line 19 ''middle covid pneumonia'' referring to risk stratification? anatomical region? please specify
  3. Line 63 Oxygen saturation on Room Air?
  4. Please provide more details upon clinical examination, any skin lesions noticed? Lung auscultation?  
  5. Line 66 please classify anaemia
  6. Line 67 Avoid using the term ''biological'' for inflammatory syndrome. When haematuria was observed, what did the urine microscopy analysis revealed? Was renal function impaired?
  7. Line 68 ''2 sets of blood cultures''
  8. Line 78 where was the pain in the left lower extremity attributed to?
  9. Please elaborate on streptococcal infection and renal manifestations
  10. Were further diagnostic tests performed in terms of basic viral panel? 
  11. Line 125 please change ''misleading''  

Author Response

Response to Reviewer 1 Comments

Dear Reviewer,

Thank you for your carefully revision of our manuscript and the relevant comments.

Answers to your comments:

Please provide specification of SARS-COV2 mode of testing (e.g line 20)

A; Specification was added.

Line 19 ''middle covid pneumonia'' referring to risk stratification? anatomical region? please specify

A: “middle pneumonia” should be referred to severity; it was changes with moderate.

  1. Line 63 Oxygen saturation on Room Air?

A: Room air was added for specification of oxygen saturation.

Please provide more details upon clinical examination, any skin lesions noticed? Lung auscultation?

A: Details were added.  

Line 66 please classify anaemia

A: Normocytic normochromic mild anaemia.

Line 67 Avoid using the term ''biological'' for inflammatory syndrome. When haematuria was observed, what did the urine microscopy analysis revealed? Was renal function impaired?

A: Haematuria was found in urine microscopy (dipstick positive). The renal function was normal.

Line 68 ''2 sets of blood cultures''

A: Revised.

Line 78 where was the pain in the left lower extremity attributed to?

A: No etiology was found for the pain of the lower limb, although we supposed to be caused by the irritation of the sciatic nerve or renal colic, possible attributed to renal involvement to the infective endocarditis.

Please elaborate on streptococcal infection and renal manifestations

A: Revised (Discussion- new paragraph).

Were further diagnostic tests performed in terms of basic viral panel?

A: Influenza virus was negative, but there were not available other tests of viral panel. 

Line 125 please change ''misleading''

  “Misleading” was changed with overlapping symptoms.

Submission Date

26 October 2021

Date of this review

09 Nov 2021 20:47:45

Reviewer 2 Report

Authors presented a case report of a previousely healthy young woman with concomitant Covid-19 infection and IE caused by S. gordonii that was treated conservatively.

The case is interesting, and describes for the first time S. gordonii IE with Covid-19 but does not provide enough back-up clinical and imaging data on causation/correlation effect of Covid-19. Honestly, it seems that initial clinical presentation of malaise and others symptoms was due to IE, and that Covid-19 was coincidental and most probably asymptomatic, and therefore probably irrelevant. 

Full insight of Covid-19 significance in this case would probably be more easily obtained if authors could provide:

  • LAB results (exact values) of IL-6, PCT, CRP, D-dimer at the time of admission
  • X ray, or event better, a MSCT thorax scan to check for Covid-19 signs of interstitial pneumonia
  • TEE images at diagnosis and at the end of antibiotic therapy
  • Information on follow-up after 3, 6 or more months

The causative link of IE and concomitant Covid-19 is not yet clear. Please refer also to a case report: "Trying to Survive A Serious Heart Condition in Time of COVID-19. Heart Surg Forum. 2021 Apr 23;24(2):E372-E374. doi: 10.1532/hsf.3815", that presents only coincidental link of Covid-19 and serious IE, with Covid-19 affecting only patients management because of delays and problem in the health care organization during the pandemic.

Author Response

Response to Reviewer 2 Comments

Dear Reviewer,

Thank you for your carefully revision of our manuscript and the relevant comments.

Answers to your comments:

  • LAB results (exact values) of IL-6, PCT, CRP, D-dimer at the time of admission

A: IL-6 was not available. The LAB results are presented in supplemental material Table A1.

  • X ray, or event better, a MSCT thorax scan to check for Covid-19 signs of interstitial pneumonia

A: MSCT was not available. Chest-X ray was not specific, with description of bilateral perihilar peribronchial thickening and interstitial infiltrates.

  • TEE images at diagnosis and at the end of antibiotic therapy

A: TEE was not performed, in the context of COVID-19 limitations in the cardiology department, but an image of TTE at diagnostic was added as supplementary material of the article (Figure A1).

  • Information on follow-up after 3, 6 or more months

A: The patient is well, cardiologic evaluation, including TEE and LAB tests after 6 months following the endocarditis was normal, but there are not available medical documents in our clinic.

The causative link of IE and concomitant Covid-19 is not yet clear. Please refer also to a case report: "Trying to Survive A Serious Heart Condition in Time of COVID-19. Heart Surg Forum. 2021 Apr 23;24(2):E372-E374. doi: 10.1532/hsf.3815", that presents only coincidental link of Covid-19 and serious IE, with Covid-19 affecting only patients management because of delays and problem in the health care organization during the pandemic.

A: New paragraph with recommended reference,

Submission Date

26 October 2021

Date of this review

11 Nov 2021 14:49:53

Reviewer 3 Report

Some (minor) language corrections are suggested here

Line 18: myalgias and weight loss.

Line 22: the presence of echocardiographic vegetations

Line 23: After 6 weeks of antibiotic treatment the patient had a favorable outcome.

Line 43: has presented

Line 71: found a mobile structure of 0.5 cm

Line 81: still occurred

Author Response

Response to Reviewer 3 Comments

Dear Reviewer,

Thank you for your carefully revision of our manuscript and the relevant comments.

Answers to your comments:

  1. Line 18: myalgias and weight loss.

A: Revised

  1. Line 22: the presence of echocardiographic vegetations

A: Revised

  1. Line 23: After 6 weeks of antibiotic treatment the patient had a favorable outcome.

A: Revised

  1. Line 43: haspresented

A: Revised

  1. Line 71: found a mobile structure of 0.5 cm

A: Revised

  1. Line 81: still occurred

A: Revised.

Submission Date

26 October 2021

Date of this review

06 Nov 2021 08:00:08

Round 2

Reviewer 2 Report

The authors have adressed all the questions raised. However, in the file invetory I can not see the image of the aortic valve beofre and after antibiotic treatment. Since this is a case of coincidental Covid-19 infection and streptococcal IE, TEE was not done due to organization and protective issues. However, TTE images should be available - but can not be downloaded.

The article meets the criteria for publication, but a case report of this kind must be accompanied by imaging - at least TTE.